# Evolution and stability of complex microbial communities driven by trade-offs

Yanqing Huang [1,2], Avik Mukherjee [1,2], Severin Schink[1], Nina Catherine Benites[1] & Markus Basan [1✉]

## Abstract

**Microbial communities are ubiquitous in nature and play an important role in ecology and human health. Cross-feeding is thought to be core to microbial communities, though it remains unclear precisely why it emerges. Why have multi-species microbial communities evolved in many contexts and what protects microbial consortia from invasion? Here, we review recent insights into the emergence and stability of coexistence in microbial communities. A particular focus is the long-term evolutionary stability of coexistence, as observed for microbial communities that spontaneously evolved in the *E. coli* long-term evolution experiment (LTEE). We analyze these findings in the context of recent work on trade-offs between competing microbial objectives, which can constitute a mechanistic basis for the emergence of coexistence. Coexisting communities, rather than monocultures of the 'fittest' single strain, can form stable endpoints of evolutionary trajectories. Hence, the emergence of coexistence might be an obligatory outcome in the evolution of microbial communities. This implies that rather than embodying fragile metastable configurations, some microbial communities can constitute formidable ecosystems that are difficult to disrupt.**

**Keywords** Coexistence; Evolution; Microbial Communities; Trade-offs
**Subject Categories** Evolution & Ecology; Microbiology, Virology & Host Pathogen Interaction

## Introduction

Diverse bacterial species have been shown to readily form interesting and complex coexisting communities in laboratory conditions (Hu et al, 2022; Sánchez et al, 2021; Martin et al, 2016; Hiltunen et al, 2017; de Visser, 2015; Levin, 1972; Hamilton and May, 1977; Vance, 1985) and natural environments (Ortiz et al, 2021; Faith et al, 2013; Deines et al, 2020). However, it is much less clear to what extent these microbial communities are stable, not only under short-term perturbations but under long-term evolutionary pressure. In other words, are these communities transient and prone to invasion and replacement by simpler microbial communities or even single strains that enter the community from the outside or may evolve from within? Or can microbial communities offer intrinsic advantages that render them 'optimal' and resistant to invasion and evolutionarily stable?

## The puzzling complexity of microbial communities

In many environments, the number of coexisting species surpasses the number of primary nutrients found in the environment. This observation appears to contradict the competitive exclusion principle, which states that different species competing for the same limited resource cannot permanently coexist (Kneitel, 2008). Based on this concept, the number of different species should be less than or equal to the number of different primary nutrients found in this environment. However, this simple prediction appears to be broadly violated, possibly in most natural communities and ecosystems. Even microbial communities that should be strongly shaped by evolutionary pressure for optimization are often more complex.

Typically, coexistence of many species in these complex communities occurs because of cross-feeding (Morris et al, 2013; Smith et al, 2019). But this does not address the questions why cross-feeding occurs at all and why it constitutes an evolutionarily stable strategy. Why should bacteria excrete valuable metabolites instead of just metabolizing them themselves? If they lack the metabolic enzymes and pathways to do so, new strains with this ability would replace the community if introduced and strains could acquire missing pathways by horizontal gene transfer. Given a limited amount of external resources, faster-growing strains would continuously increase in abundance at the expense of slower-growing strains, unless there are stabilizing interactions. Consider the case where a fast-growing strain produces fermentation products that slower-growing strains can utilize. Even in this case, slower-growing strains that grow on these fermentation products would be at a disadvantage if the faster-growing strain can utilize the fermentation product equally well once the primary nutrient is depleted. In this case, the population fraction of the faster-growing strain would increase with every growth cycle until the slower-growing strain is eliminated from the population. Therefore, the strain specializing in the fermentation product must have some intrinsic metabolic advantage in utilizing the fermentation product. But what are these advantages and why should they exist in the first place?

According to both of these scenarios, complex microbial communities, based on cross-feeding, should be merely transient states, whereas evolution ultimately favors less complex systems

[1]Harvard Medical School, Department of Systems Biology, Boston, USA. [2]These authors contributed equally: Yanqing Huang, Avik Mukherjee. ✉E-mail: markus@hms.harvard.edu

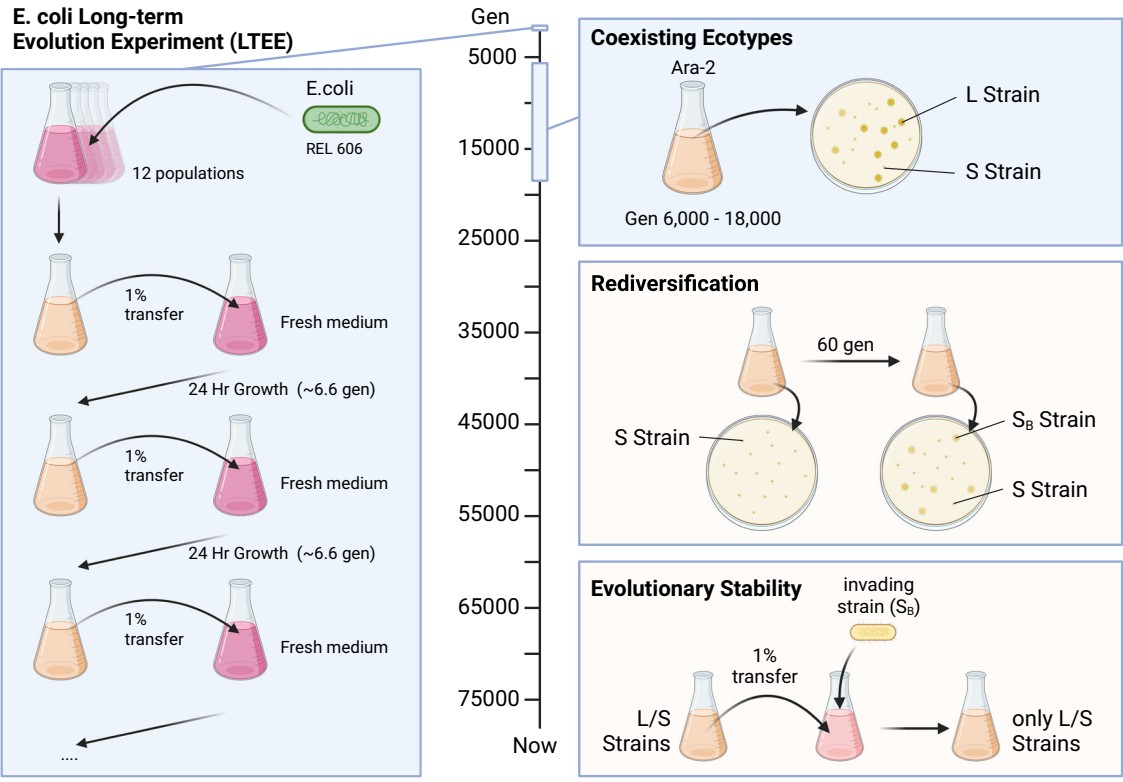

**Figure 1. Schematic of LTEE coexistence and rediversification experiments.**

Left panel: LTEE experiment set up. Cells were grown in minimal medium containing a limiting amount of glucose (~138 μM) and citrate. Each day, 1% inoculum was transferred to fresh media. Right panel, top: phenotypic diversification in LTEE and emergence of L and S strains. L strains form larger colonies and S strains form smaller colonies. Middle: S train monoculture evolved a L-like strain, $S_B$, as reported by the Hallatschek lab (Ascensao et al, 2024). Bottom: The $S_B$ strain is outcompeted in a L/S coculture, and stable coexistence of L and S strain is maintained (Ascensao et al, 2024).

with fewer species that have acquired the ability to completely utilize available resources. Laboratory evolution experiments have put this theoretical expectation to the test, with unexpected outcomes.

## Evolution and stability of coexistence

### Coexisting ecotypes in the *E. coli* long-term evolution experiment

In stark contrast with these expectations, the *E. coli* long-term evolution experiment (LTEE) pioneered by the lab of Richard Lenski experimentally demonstrated that evolution can favor increasing complexity, even in simple, well-mixed environments (Lenski et al, 1991; Fitch et al, 1994). Despite the simplicity of the LTEE growth environment, the experiment that started with a single lab strain and resulted in the emergence of complex ecological communities consisting of multiple strains and complex interactions, as illustrated in Fig. 1 (Rozen and Lenski, 2000; Rozen et al, 2005). Coexistence between multiple strains was unexpected and discovered by accident based on colony morphology, where one strain was denoted L-strain (large colonies) and the other one S-strain (small colonies). Moreover, the L- and S-strain community persisted for thousands of generations (Rozen and Lenski, 2000;

Rozen et al, 2005). Even as the L- and S-strains continuously evolved, each sub-population retained key phenotypic similarities (Rozen et al, 2005). These observations suggest that interactions between the L- and S-strains render this simple community stable under continuous evolution.

The advent of sequencing technology revealed an abundance of coexistence phenomena across 60,000 generations of the long-term evolution experiment (Good et al, 2017). By measuring allele frequency changes, long-term coexistence was found in most of the 12 separate initial populations. Unlike the L/S strains, a phenotypic characterization of other coexisting LTEE strains is largely lacking. However, recurrent genetic mutations were found in many of these evolved strains, suggesting that similar mechanisms of coexistence could have resulted in these communities (Deatherage and Barrick, 2021).

### Rediversification after removal of one of the ecotypes

Recently, the lab of Oskar Hallatschek added another important piece of this puzzle (Ascensao et al, 2024) by demonstrating rediversification after removal of one of the L/S ecotypes. Remarkably, they found that strains separated by 30,000 generations of evolution rediversified in a similar manner (Fig. 1, right panel). Key phenotypes of the original community quickly reemerged, including differences in lag times and starvation

survival between the coexisting strains, though mutation and genetic changes that resulted in these phenotypes and affected the coexistence were highly varied (Ascensao et al, 2024, 2023). This suggests that rather than the conservation of specific mutations, as might be expected, there are many alternative routes how genetic changes can result in the same phenotypic changes. This work further highlights the importance of phenotypic convergence as the basis of evolution of coexistence. Laboratory experiments in rediversification can help explain the wide range of recovery phenomenon in the wild after sudden environmental shifts cause the removal of ecotypes (Rozen et al, 2009). Thus, an emergent theme is the conservation of specific phenotypes that ultimately form the basis of interactions, resulting in coexisting communities.

## Short-term stability of coexistence

What are the mechanisms that underpin the L/S-strain coexistence? Early work by Rozen and Lenski demonstrated that coexistence of the L-strain and S-strain is based on acetate excretion of the L-strain and a higher fitness of the S-strain after glucose depletion based on acetate utilization (Rozen and Lenski, 2000). It is noteworthy to mention, that Lenski and colleagues used limiting amount of glucose (~138 μM) in the minimal media and thus allowed a diauxic shift to happen naturally in every passage. In LTEE, cultures stop growing and reach a starvation-like state as nutrients are used up but never reach a stationary phase due to excess culture density. This results in coexistence at stable frequencies of the two strains: An excess abundance of the L-strain results in higher acetate production that helps the S-strain recover. Conversely, high S-strain frequencies allow the faster growing L-strain population to recover during the growth phase on glucose. Hence, this interplay can explain the stability of the coexistence on short timescales without the introduction of new strains. However, these arguments do not explain why the L/S-strain community is resistant to invasion and not elimination on long timescales? In other words, why did a fast-growing strain that does not excrete acetate or one that is more adept at utilizing acetate after glucose depletion not manage to invade the ecosystem? If the emergence of the L/S-strain phenotypic community is not an evolutionary accident, as its persistence in the LTEE and its recurrence after removal of one strain suggests, the mystery of evolutionary stability must be addressed.

## Evolutionary stability of coexistence

We argue that the answer to this question is rooted in physiological and biochemical constraints of central carbon metabolism that promote coexistence of two strains based on interacting trade-offs and at the same time prohibit invasion of the community by a third invader. Our lab has recently shown using simulations of the LTEE that the interaction of two obligatory trade-offs can result in convergent evolution of stable coexistence between two strains that closely resemble the L/S-strain phenotypic community (Mukherjee et al, 2023). The first trade-off mandates that faster growth result in higher acetate excretion and has been shown to be rooted in the protein cost of energy production of fermentation versus respiration (Basan et al, 2015). This trade-off prohibits the existence of a fast-growing strain that does not produce acetate. The second trade-off is between fast growth and short lag times when switching

to acetate utilization after glucose depletion (Basan et al, 2020) and has been shown to result from the irreversibility of metabolic reactions in the central carbon metabolism (Schink et al, 2022b; Basan et al, 2020). This trade-off prohibits a fast-growing strain from quickly switching to acetate after the diauxic shift and benefitting from the acetate excreted into the medium. It had been previously shown that this trade-off can result in unexpected coexistence in multi-resource environments of different bacterial species (Bloxham et al, 2022).

## Simulation of evolutionary dynamics reveals phenotypic convergence

Using evolutionary simulations, constrained by these two trade-offs, we showed that any combination of randomized initial strains converges to the same stable phenotypic strain combination (Mukherjee et al, 2023). We also showed that any additional strain, lacking an advantage in either trade-off, is quickly outcompeted. At the same time, new coexisting strains can emerge, but phenocopy the L and S strains, precisely as observed in the LTEE and as observed by Ascensao et al (Ascensao et al, 2024). The phenotypic combination of this stable fixed point in the simulation qualitatively matched phenotypes of the L- and S-strains. Our model further predicted that coexisting strains could start with different initial growth rates, but would ultimately converge to the same growth rate combination.

## Interaction of multiple trade-offs is required for evolutionary stability

It is important to point out that multiple trade-offs are required for the evolutionary stability of L/S-strain community and to prevent invasion. Starvation survival has been proposed to play a role in L/S-strain coexistence (Vasi and Lenski, 1999; Lenski and Velicer, 2001) and another trade-off between fast growth and starvation survival has recently been uncovered (Biselli et al, 2020; Schink et al, 2022a, 2024), where faster growth leads to impaired starvation survival. However, this growth-starvation trade-off alone does not suffice to explain the evolutionary stability of L/S-strain coexistence if the improved survival of the S-strain were not coupled with the availability of acetate in the medium and unless the S-strain had some kind of advantage for using acetate for its survival. Otherwise, a single strain with the optimal combination of growth rate and starvation survival would simply take over the population over time and break the coexistence. Thus, the coupling of several trade-offs via acetate excretion into the medium is essential for the stability of L/S-community, as a combination of distinct strains can take advantage of the fast growth phase on glucose and utilize the acetate that is being secreted during this phase, better than any single strain constrained by these trade-offs could.

## Evolutionarily stable communities can select for sub-maximum growth rate

The L/S-strain coexistence is a situation in which evolution favors slow, sub-maximum growth for at least one strain. This occurred on glucose in the LTEE, which is thought to be one of the 'best' substrates of *E. coli* and nevertheless, resulted in slower growth of

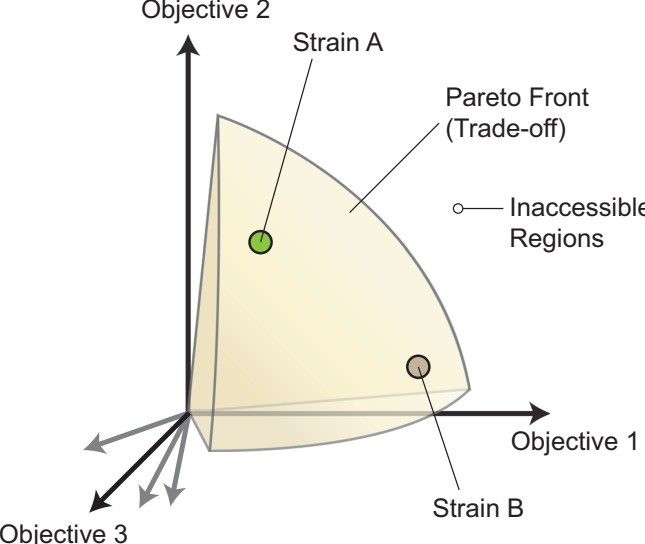

**Figure 2. Hyperspace of phenotypic objectives and resultant trade-off space.**

The Pareto front separates an accessible region of phenotype space from an inaccessible region that is prohibited by metabolic, biochemical, or biophysical constraints. Strains can exist within the hypervolume confined by the Pareto hypersurface. On the Pareto front, one phenotypic objective can only be improved by decreasing others.

the S-strain. Could such interactions within microbial communities have shaped growth rates of naturally occurring wildtype strains on different substrates?

Lab culture conditions in the LTEE are hardly typical of natural bacterial environments and how common is this situation in natural environments? Typically, the large range of growth rates of bacteria on different substrates are thought to reflect intrinsic properties of these substrates such as energy content (Neidhardt, 1996), protein cost of catabolizing them (Hui et al, 2015; Scott et al, 2010), or limitations in membrane capacity for their uptake (Zhuang et al, 2011). However, different bacterial species grow at vastly different rates on the same substrates. Glucose is a supposedly rich substrate for *E. coli*, but it appears to be a rather poor substrate for *P. aeruginosa* (Schink et al, 2022b), which casts doubt on the notion of carbon quality as a fundamental biochemical property. Moreover, various works have shown that growth rates of *E. coli* on many supposedly 'poor' substrates can be improved (Pettigrew et al, 1996; Applebee et al, 2011; Basan et al, 2017, 2020; Zwaig and Lin, 1966), sometimes dramatically, simply by swapping promoters of carbon specific operons and metabolic enzymes (Mukherjee et al, 2024). Swapped growth rates resulted in a swapping of a host of other phenotypes that may trade-off against growth rate. These findings suggest that growth laws that have been found to govern an interplay of ribosome content, gene expression and growth rate (Scott et al, 2010; Scott and Hwa, 2011) could be the result of a simple regulatory architecture, mediated by cAMP and ppGpp (You et al, 2013; Kochanowski et al, 2021), serving in part, to implement a set of trade-offs between growth rate and other cellular capabilities, such as fast switching between substrates, motility, starvation, and stress survival (Mukherjee et al, 2024).

The plasticity of bacterial growth laws that encode these phenotypic trade-offs (Mukherjee et al, 2024) may explain how it is possible that the L/S-strain coexistence reemerged so quickly after removal of one strain. In this picture, rather than acting as a fundamental biochemical limitation caused by the substrate, growth rates constitute a regulatory knob that can be quickly dialed by evolution to strike the optimal balance between fast growth and adaptability and optimize associated trade-offs. Rather than being a side-effect of resource allocation at different growth rates due to intrinsic substrate quality (Liu et al, 2005), trade-offs can be the driving force that determines microbial growth rates and the LTEE suggests that this may frequently occur in the context of microbial communities (Rozen and Lenski, 2000; Rozen et al, 2005).

## Trade-off space

Based on these findings, we can summarize that coexistence of bacterial communities can be caused by the interaction of multiple phenotypic trade-offs, rooted in biochemical and metabolic constraints. For each strain, we can visualize its combination of phenotypes as a distinct point in the high-dimensional space of phenotypic objectives (Fig. 2). As discussed in this review, improving one phenotypic objective (such as fast growth), often comes as a cost of decreasing its ability to optimize on another objective (such minimizing acetate excretion, minimizing lag time or maximizing starvation survival). These constraints create a Pareto front that represent optimal phenotypic combinations. On the Pareto front, any phenotypic objective can only be increased by decreasing another. The Pareto front separates a space of suboptimal phenotypic combinations, where objectives can be further improved, from an inaccessible regime that is not permitted by biochemical or physical constraints. Evolution will drive strains toward the Pareto front and strains that fall into the suboptimal phenotype space can be outcompeted. Thus, for simplicity, in our discussion we assume that all strains fall directly on the Pareto front.

Note that two strains that are close neighbors in this trade-off space cannot coexist as they occupy the same phenotypic niche. However, as shown in our simulations (Mukherjee et al, 2023), two distant strains in phenotypic space can coexist in the same environment, but only if one strain creates an ecological niche for the other that it cannot itself occupy. For example, the phenotypic trade-offs between growth rate, fermentation rate, and lag time result in coexistence. Fast growth requires obligatory excretion of the fermentation product acetate, but fast growth also results in long lag times for acetate utilization. This creates an ecological niche for a slower-growing strain that utilizes acetate more quickly. There is a particular combination of strains in this phenotype space that is optimal and thus cannot be invaded by any other strain.

To illustrate the possible trade-offs that can emerge from competing phenotypic objectives, we have grouped trade-offs in six categories, as depicted in Fig. 3.

## Obligatory trophic interactions

Fundamental trade-offs that are based on basic physiological constraints such as proteome allocation, enzymatic reaction rates,

# Types of Trade-offs

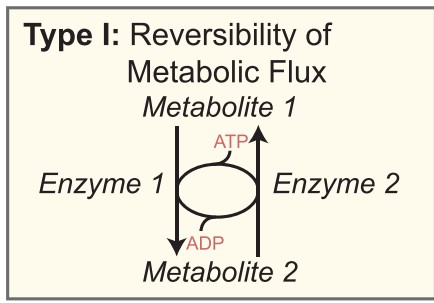

**Type I:** Reversibility of Metabolic Flux

*Metabolite 1*

*Enzyme 1* | *Enzyme 2*

ATP | ADP

*Metabolite 2*

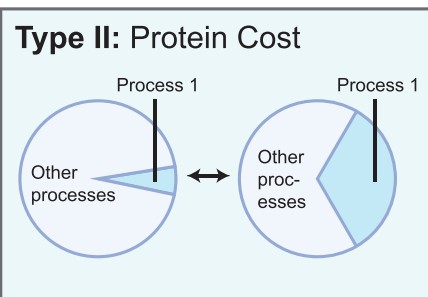

**Type II:** Protein Cost

Process 1 | Process 1

Other processes | Other processes

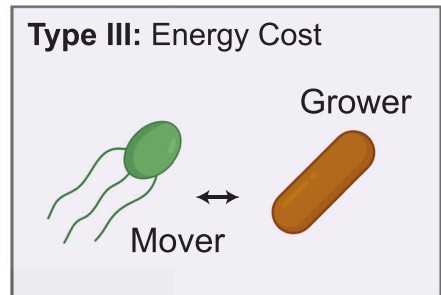

**Type III:** Energy Cost

Grower

Mover

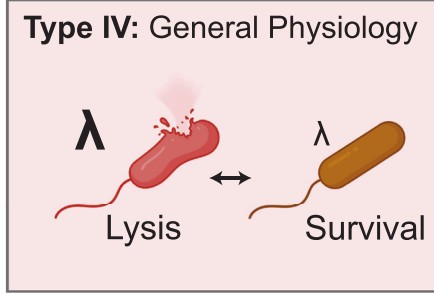

**Type IV:** General Physiology

λ | λ

Lysis | Survival

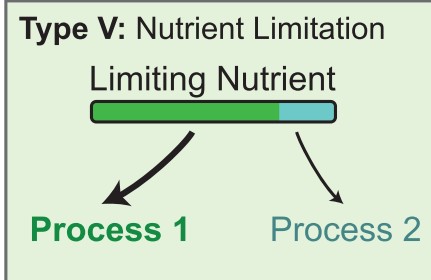

**Type V:** Nutrient Limitation

Limiting Nutrient

**Process 1** | Process 2

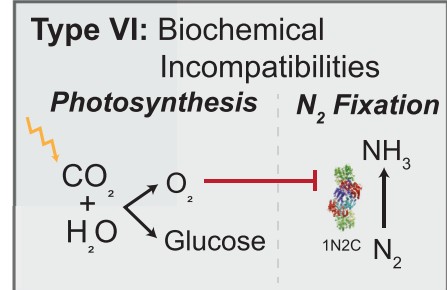

**Type VI:** Biochemical Incompatibilities

*Photosynthesis* | *N₂ Fixation*

$CO_2 + H_2O$ → $O_2$ → Glucose ⊣ $NH_3$ ← 1N2C ← $N_2$

**Figure 3.  Classification of six categories of trade-offs.**

Type I: Reversal of metabolic flux between some essential metabolites is not achieved via reversibility of the same set of enzymes. Instead, opposing directions are achieved by different chemical reactions catalyzed by different enzymes. This creates an intrinsic trade-off between fluxes in opposing directions and the amount of futile cycling taking place in the system, which dissipates ATP (Schink et al, 2022b; Basan et al, 2020). Type II: Proteins constitute the central machinery of most biological processes, but proteomic resources are constrained, partly because protein production itself requires proteomic resources in the form of ribosomes. Therefore, there is a basic resource allocation trade-off between all biological process in the cell. Investing a larger fraction of the proteome in one process results in an obligatory decrease in the investment in other processes. Type III: In certain situations, such as in substrate starvation, cells can be ATP-limited and cannot simply produce more ATP by investing more proteome into energy production pathways. In such situations, there is a trade-off between different energy consuming processes. Type IV: One could imagine trade-offs between general physiological parameters. For example, faster growing cells exhibit higher cell envelope permeability that results in impaired starvation survival (Schink et al, 2024). Type V: If any nutrient or micronutrient is limiting, reflecting a chemostat-like environment, this results in a trade-off between different biological processes that have different requirements for this limiting nutrient. Type VI: There can be fundamental incompatibility between different biochemical processes. For instance, a side product of one process, or its required conditions, may interfere with another, resulting in a trade-off between the two processes. An example of this trade-off, with wide-ranging ecological consequences results from the oxygen-sensitivity of nitrogenase enzymes (Gallon, 1981). This results in incompatibility of two central biochemical processes of life on earth: nitrogen fixation and photosynthesis. Photosynthesis provides ATP, which is required for nitrogen fixation, but produces oxygen as a waste product. Oxygen inhibits nitrogenase, but fixed nitrogen is required for biosynthesis of all proteins, including the photosynthetic machinery. This trade-off causes either spatial or temporal separation of these important processes and results in frequent symbiotic coexistence of nitrogen-fixing and photosynthetic organisms.

and reversibility of reactions in central metabolism, should be conserved between strains and not easily overcome during long-term evolution. Therefore, we believe such trade-offs may underlie coexistence in a variety of natural microbial systems. In fact, 'Trophic interactions' are thought to be core to microbial communities, i.e., molecules excreted by one species form the primary nutrient of other species (Gralka et al, 2020). Acetate excretion by the L-strain, and consumption by the S-strain would constitute an obligatory trophic interaction, resulting in stability of the ecosystem. Similarly, we believe that other trophic interactions could be rooted in physiological constraints, too. The succession of marine bacteria degrading organic matter in the ocean requires excretion of nutrients for stability (Datta et al, 2016). Trophic interactions have been extensively studied in the human gut microbiome (Wang et al, 2019; Fischbach and Sonnenburg, 2011), and have numerous applications for synthetic biology. For example, the coexistence of Firmicutes and Bacteroidetes in the human gut, requires excretion of short-chain fatty acid for stability (Cremer

et al, 2017). Trade-offs based on cellular crowding and spatial organization have also been shown to result in coexistence in evolution of laboratory yeast populations (Frenkel et al, 2015).

## Acid-stress induced metabolite exchange can lead to coexistence

Recently, a new mechanism that results in cross-feeding and coexistence has been uncovered (Amarnath et al, 2023). Amarnath et al showed that acid stress, resulting from high-levels acetate excretion can trigger cross-feeding, where "sugar-eating" strains excrete metabolites such as pyruvate and lactate that enables "acid-eating" strains to detoxify the medium and growth to resume. This work suggest that microbial communities pass through periodic phases of growth and collaborative deacidification. The stability of coexistence in these conditions emerges because of the division of microbes into those with preferential glycolytic metabolism and those with a preferentially gluconeogenic metabolism (Schink et al, 2022b; Gralka

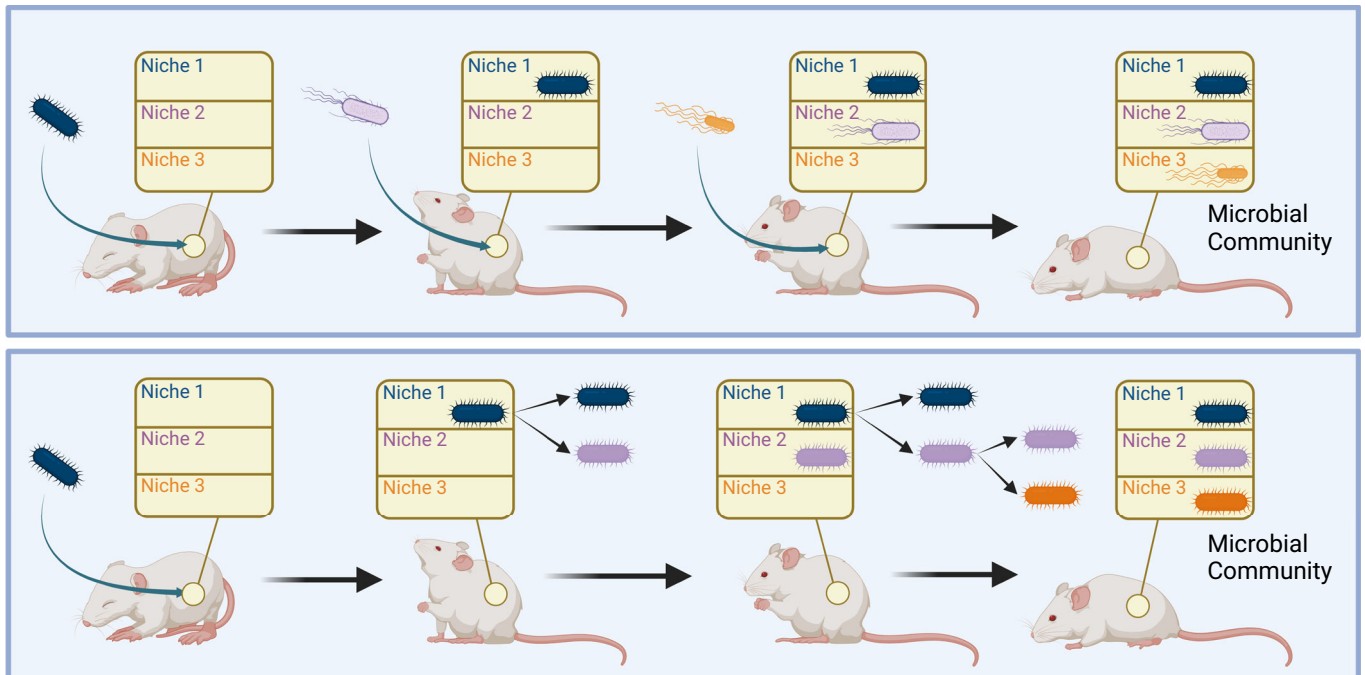

**Figure 4.  Modalities of microbiome establishment: colonization (top panel) versus diversification (bottom panel) in the assembly of microbiota.**

Complex microbiota are typically thought to assemble with strains taken up from the environment. However, it is also possible that just like the emergence of ecotypes in the LTEE, individual strains may evolve and diversify into distinct lineages to occupy different ecological niches. While there is some evidence of this diversification process in human gut microbiota (Zhao et al, 2019; Barreto and Gordo, 2023), its overall role in the assembly of microbiota is an open question.

et al, 2023). This trade-off between glycolysis and gluconeogenesis preference is rooted in biochemical constraints of central metabolism and the reversibility of metabolic flux (Schink et al, 2022b).

## Spatial structure and stability of coexisting communities

Spatial structure has long been proposed as a stabilizing factor for microbial communities (Gauze and Gauze, 1934; Brauchli et al, 1999; Amarasekare, 2003; Chesson, 2000; Durrett and Levin, 1998; Lobanov et al, 2023; Hyun et al, 2008). Many natural environments of microbial communities are spatially structured, like mucosa-associated microbiota (Tropini et al, 2017; Sonnenburg et al, 2004; Thursby and Juge, 2017; Juge, 2022), biofilms found in natural environments (Yang et al, 2011), and even cultivated microbial communities, such as kefir (Blasche et al, 2021). Spatial structure is also linked to cell motility, as some strains are fast-moving ("mover") in response to environmental stimuli (i.e., nutrient sources) whereas other strains are slow-moving but fast growing ("grower") (Gude et al, 2020). These interactions can be traced back to energy-based trade-offs or proteome-based trade-offs, as cell motility is proportional to energy expenditure and proteome investment. Chemotaxis in combination with such trade-offs can result in coexistence in temporally varying, spatially non-homogeneous environments (Keegstra et al, 2022).

## Diversification and de novo emergence of coexistence in the gut

The question emerges if the spontaneous evolution of coexisting communities in the Lenski LTEE is specific to this simple and

controlled growth environment or if similar phenomena play a role in natural environments and contribute to the assembly of complex ecosystems. Gut microbiota are microbial communities that have attracted considerable interest in recent years due to their relevance to human health and disease (Nicholson et al, 2012; Lozupone et al, 2012; Tamburini et al, 2018; Monira et al, 2017; Faith et al, 2011; Filippo et al, 2010). These exceedingly complex microbial communities are typically thought to assemble by colonization with strains in the environment, including transmission from the mother (Barreto and Gordo, 2023). However, it is also plausible that after initial colonization, individual strains diversify to occupy different ecological niches, like the emergence of coexisting communities in the LTEE. In fact, it has recently been observed in healthy humans that after colonization with single strains of *Bacteroides fragilis*, these strains diversified into coexisting lineages (Zhao et al, 2019; Barreto and Gordo, 2023). The same genes were repeatedly mutated across different individuals in this diversification process, suggesting an adaptive role of these mutations. These observations fall within a broader body of work, demonstrating rich dynamics of interhost evolution of gut microbiota, recently reviewed by Barreto and Gordo (Barreto and Gordo, 2023). So far, these are just isolated observations, but it seems possible that spontaneous diversification into coexisting communities is quite common in the assembly of complex microbiota, as illustrated in Fig. 4. Lessons learned from the emergence of coexistence in the LTEE might be applicable to natural microbial communities and ecosystems. One of these lessons is that to understand the underlying reasons for emergence and stability of coexistence, it may be necessary to go beyond genetic changes and identify recurrent phenotypes. Such phenotypes can give hints regarding underlying biochemical

constraints and trade-offs. It may even be possible that very different microbial species can occupy the same ecological niche in different communities by having evolved similar phenotypic profiles.

## Conclusions and future perspectives

Coexistence can emerge from different mechanisms ranging from cross-feeding to spatially heterogeneous environments. However, such patterns are not sufficient to explain the stability of coexisting communities on long timescales under continuing evolutionary pressure. In other words, the question arises why a single strain is unable to successfully occupy these niches, potentially with some spatiotemporal physiological adaptation. We argue that ultimately, the reasons for specialization must be rooted in physical, biochemical, and metabolic constraints that are not easily overcome by evolution. Therefore, to understand the frequent emergence and long-term stability of microbial communities, a more molecular approach is required to elucidate the underlying trade-offs. This may also lead to a more mechanistic understanding of microbial communities. Phenotypic characterization may reveal another important piece of this puzzle. It is likely that different genetic mutants, different strains, or even different microbial species occupy identical phenotypic niches within microbial communities, greatly reducing the complexity of these systems, but only a careful phenotypic characterization of the strains in these communities can reveal such patterns.

Such an approach would constitute a remarkable convergence of different fields, ranging from laboratory evolution, to microbial ecology, molecular physiology, and metabolism. Perhaps this should not be surprising given that "nothing in biology makes sense except in the light of evolution," as stated by Theodosius Dobzhansky in a 1973 essay (Dobzhansky, 1973). We argue that this also means that to understand evolution, it is essential to uncover the molecular principles and trade-offs that constrain the phenotypic landscape on which evolution takes place.

## Peer review information

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

## Acknowledgements

This project was supported by MIRA grant (5R35GM137895) and an HMS Junior Faculty Armenise grant by Giovanni Armenise-Harvard Foundation (GAHF) to MB. NCB was supported by following grants, National Science Foundation Graduate Research Fellowship Program (DGE 2140743) and Systems, Synthetic, and Quantitative Biology Training grant award (T32GM135014). Any opinions, findings, and conclusions or recommendations expressed in this material are those of the author(s) and do not necessarily reflect the views of the National Science Foundation.

## Author contributions

**Yanqing Huang**: Conceptualization; Visualization; Methodology; Writing—original draft; Writing—review and editing. **Avik Mukherjee**: Conceptualization; Visualization; Methodology; Writing—original draft; Writing—review and editing. **Severin Schink**: Methodology; Writing—review and editing. **Nina Catherine Benites**: Writing—review and editing. **Markus Basan**: Conceptualization; Supervision; Funding acquisition; Visualization; Writing—original draft; Project administration; Writing—review and editing.

## Disclosure and competing interests statement

The authors declare no competing interests.

