## [Peer Review File · Molecular Systems Biology]

Evolution and stability of complex microbial communities driven by trade-offs

Yanqing Huang, Avik Mukherjee, Severin Schink, Nina Benites, and Markus Basan

Corresponding author(s): Markus Basan (markus@hms.harvard.edu)

Review Timeline:

Editorial Decision:	17th Jun 24
Revision Received:	18th Jun 24
Accepted:	20th Jun 24

Editors: Maria Polychronidou and Jingyi Hou

Transaction Report:

17th Jun 2024

Manuscript Number: MSB-2024-12387

Title: Evolution and stability of complex microbial communities driven by trade-offs

Author: Yanqing Huang

Avik Mukherjee

Severin Schink

Nina Benites

Markus Basan

Dear Markus,

Thank you again for submitting this Review Article. We have now heard back from the two reviewers who agreed to evaluate your manuscript. As you will see from the reports below, the reviewers provide enthusiastic support and think that the manuscript is interesting and relevant. They only mentioned a few minor issues that we would ask you to address in a revision of the current manuscript.

Additionally, on a more editorial level, please do the following:

Please click on the link below to submit your revised paper.

I look forward to receiving your revised manuscript soon.

Kind regards,
Jingyi

Jingyi Hou, PhD
Scientific Editor
Molecular Systems Biology

Reviewer #1:

I found this to be a compelling review of recent results highlighting the importance of bacterial physiology in understanding the remarkable degree of coexistence found within microbial communities (and even initially clonal populations evolving as in the case of the LTEE). I believe that the manuscript can be published essentially as is (but there were a few grammatical issues / typos, some of which are pointed out below).

One thing that I have been struck by is the different priors that researchers come to this issue with. Microbial evolution researchers focus on growth rate, with the strong prior that the faster grower will drive the slower grower (often ancestral strain) extinct. This of course does happen, but only when the two strains are occupying the same "niche". However, in many cases there are ecological interactions that emerge and lead to coexistence (as in the case of cooperator-cheater strains when the cooperator has preferential access to the fruit of its labor). This is one way to think about the emergence of the stable lineages in the LTEE. I also think that it is powerful to highlight that evolution continues in each lineage, but if there is a selective sweep it occurs within only one sub-population. Indeed, this is an intriguing mechanism for speciation potentially (even without spatial separation).

A few minor points:

"Even as the L- and S-strains continuously evolved, they retained key phenotypic similarities (Rozen et al, 2005)." Wasn't clear to me whether the authors are referring to similarities between the L and S strains or instead to persistent phenotypes within each sub-population during evolution.

"The advent of sequencing technology revealed an abundance of coexistence phenomenon across 60,000 generations of the long-term evolution experiment (Good et al, 2017)." -> phenomena

"This trade-off prohibits a fast-growing strain from quickly switch to acetate after the diauxic..."  switching

Reviewer #2:

In "Evolution and stability of complex microbial communities driven by trade-offs", the authors review recent insights into the emergence and stability of coexistence in microbial communities.

The review is timely and well written, providing an enjoyable read. It comes from a group that made important contributions in the field of bacterial physiology. The authors argue that the mechanistic basis of stability and coexistence in microbial communities is still not fully understood. They draw heavily on their findings from the E. coli long term evolution experiment (LTEE) to integrate insights with recent theoretical and empirical work on trade-offs in microbial metabolism and ecology.

The review covers a wide range of recent studies, from the LTEE to theoretical models, addressing microbial community dynamics comprehensively. The manuscript is logically structured and easy to follow, systematically addressing different aspects of microbial community stability, from short-term interactions to long-term evolutionary dynamics. By focusing on trade-offs, the authors provide a potential mechanistic framework applicable to microbial communities in various contexts.

While the review discusses trade-offs extensively, it sometimes lacks mechanistic insights into how these tradeoffs are regulated at the molecular level. Enhancing the discussion by providing examples of mechanisms underlying the specific types of trade-offs depicted in Figure 3 could improve the review.

The authors could also expand on the future perspectives with concrete questions. While the phenotypic space presents a promising framework for understanding microbial communities, it would be beneficial to outline specific questions and phenotypes that are within reach of current experimental and theoretical approaches.

Specific comments:

1. Reference check: Verify if the reference below corresponds to the recently published paper on Nature Physics (<https://doi.org/10.1038/s41567-024-02511-2>)

Schink S, Polk M, Athaide E, Mukherjee A, Ammar C, Liu X, Oh S, Chang Y-F & Basan M (2022b)
The energy requirements of ion homeostasis determine the lifespan of starving bacteria.
bioRxiv: 2021.11.22.469587

2. Fig. 3 (Type II panel): The pie chart chunk devoted to Process 1 in the second pie chart could be made larger to improve clarity.

3. Fig. 3 (Type V panel): The figure currently does not clearly convey the idea of a trade-off. Consider different ways to improve this panel, such as making one line corresponding to the limiting nutrient and breaking it into two pieces with different colors representing processes 1 and 2.

4. Figure 4: There is no citation to Figure 4 in the text despite a section dedicated to it. Additionally, the caption of Figure 4 does not explain the different panels. Adding references to the upper and lower panels in the caption (e.g., in '..just like the emergence of ecotypes.. (lower panel)') could enhance clarity.

Overall, this review is timely, well written, and covers the relevant recent literature on the topic of stability and emergence of microbial communities, bringing in a physiological angle. By addressing the suggested improvements the authors could make it even clearer.

All editorial and formatting issues were resolved by the authors.

20th Jun 2024

Manuscript number: MSB-2024-12387R

Title: Evolution and stability of complex microbial communities driven by trade-offs

Dear Markus,

Thank you again for sending us your revised manuscript. We are now satisfied with the modifications made and I am pleased to inform you that your paper has been accepted for publication.

Thank you for your contribution to Molecular Systems Biology.

Best wishes,
Jingyi

Jingyi Hou, PhD
Scientific Editor
Molecular Systems Biology
